# Epithelial Ovarian Cancer: A Five Year Review

**DOI:** 10.3390/medicina59071183

**Published:** 2023-06-21

**Authors:** Christos Arnaoutoglou, Kalliopi Dampala, Christos Anthoulakis, Evangelos G. Papanikolaou, Ioannis Tentas, Georgios Dragoutsos, Nikolaos Machairiotis, Paul Zarogoulidis, Aristeidis Ioannidis, Dimitris Matthaios, Eleni I. Perdikouri, Dimitrios Giannakidis, Chrysanthi Sardeli, Stamatios Petousis, Panagoula Oikonomou, Christina Nikolaou, Charalampos Charalampidis, Konstantinos Sapalidis

**Affiliations:** 11st Department of Obstetrics & Gynecology, Papageorgiou Hospital, Aristotle University of Thessaloniki, 541 24 Thessaloniki, Greece; arnaoutoglou7@gmail.com (C.A.); dampalakalliopi@hotmail.com (K.D.); christos.anthoulakis@hotmail.com (C.A.); 23rd Department of Obstetrics & Gynecology, Hippokration Hospital, Aristotle University of Thessaloniki, 541 24 Thessaloniki, Greece; papanikolaous@assistingnature.gr; 3Department of Obstetrics & Gynecology, General Hospital of Giannitsa, 581 00 Giannitsa, Greece; j.tentas@gmail.com; 4Department of Obstetrics and Gynecology, Democritus University of Thrace, 681 00 Alexandroupolis, Greece; geodragoutsos@gmail.com; 5Fellow in Endometriosis and Minimal Access Surgery, Northwick Park, Central Middlesex and Ealing Hospitals, London North West University Heathcare, NHS Trust, London NW10 7NS, UK; nikolaosmachairiotis@yahoo.com; 63rd University General Hospital, “AHEPA” University Hospital, 546 36 Thessaloniki, Greece; sapalidiskonstantinos@gmail.com; 7Surgery Department, Genesis Private Hospital, 546 26 Thessaloniki, Greece; ariioann@yahoo.gr; 8Oncology Department, General Hospital of Rhodes, 851 33 Rodos, Greece; dimalexpoli@yahoo.com; 9Oncology Department, General Hospital of Volos, 382 22 Volos, Greece; eigrouse@yahoo.gr; 101st Department of Surgery, Attica General Hospital “Sismanogleio-Amalia Fleming”, 151 26 Athens, Greece; giannakidis.d@gmail.com; 11Department of Pharmacology & Clinical Pharmacology, School of Medicine, Aristotle University of Thessaloniki, 541 24 Thessaloniki, Greece; sardeli@auth.gr; 122nd Department of Obstetrics and Gynaecology, Aristotle University of Thessaloniki, 541 24 Thessaloniki, Greece; petousisstamatios@gmail.com; 13Surgery Department, Democritus University of Thrace, 691 00 Alexandroupolis, Greece; pen.ek@hotmail.com (P.O.); cnikolaou3@gmail.com (C.N.); 14Pathology Department, University of Cyprus, Nicosia 1678, Cyprus

**Keywords:** ovarian cancer, local treatment, cancer, gene therapy, prognostic factors, ovarian cancer

## Abstract

Ovarian cancer is a malignant disease that affects thousands of patients every year. Currently, we use surgical techniques for early-stage cancer and chemotherapy treatment combinations for advanced stage cancer. Several novel therapies are currently being investigated, with gene therapy and stem cell therapy being the corner stone of this investigation. We conducted a thorough search on PubMed and gathered up-to-date information regarding epithelial ovarian cancer therapies. We present, in the current review, all novel treatments that were investigated in this field over the past five years, with a particular focus on local treatment.

## 1. Introduction

Ovarian cancer is the third most common cancer among females after breast and lung cancer [1]. There were 20,000 new patients and 13,000 thousand deaths every year, according to the SEER cancer statistics review, in the past 40 years [2]. There are novel surgical techniques for ovarian cancer and several new drugs that increase the five-year overall survival rate; however, the five-year post-diagnosis survival rate is still less than 50% [3,4,5]. Currently, several biomarkers are being investigated for every disease diagnosis [6]. Breast Cancer 1 (BRCA1) and Breast Cancer 2 (BRCA2) pathways are observed in over 18% of women with High-Grade Serous Ovarian Cancer (HGSOC). Breast Cancer 1 (BRCA1) and Breast Cancer 2 oncogenes are inherited as mutations associated with breast cancer. The Breast Cancer 1 and Breast Cancer 2 genes are included on chromosomes 17q21 and 13q12, respectively. MAPK/ERK Pathway is observed in ovarian cancer. The MAPK/ERK pathway promotes the invasion of the disease and induces metastasis and chemotherapy resistance. It was previously shown that epidermal growth factor receptor (EGFR) is overexpressed in 75% of ovarian cancers. The EGFR/AKT signalling pathway and epidermal growth factor and tumor growth factor activate the promotion and inhibition of tumour survival. Integrins are receptors expressed on the cell surface that consist of two subunits: subunit α and subunit β. The integrin inhibitors are considered as therapeutic agents for ovarian carcinoma. Moreover, GRP78 molecule is considered to be a delivery system for ovarian cancer drugs. The p38alpha, which are small compound inhibitors that have been administered in clinical trials, block p38 genes to reduce the production of ovarian cancer cells [7].

There are still several unknown factors associated with the disease; therefore, we still need several aspects of the disease pathways to be elucidated at a cellular level. The number of metastases is a clinical prognostic factor for patients with epithelial ovarian cancer [8,9,10]. Previous studies presented data relating the number of sites-distant metastases at diagnosis with survival. Moreover, there are still not enough data for large populations regarding the estimation of prognosis for newly diagnosed epithelial ovarian cancer patients with several distant metastases. Metastasis in liver, bone, lung, and/or brain was previously observed in autopsy studies, despite not being apparent prior to death [11,12]. Further studies on both localized and/or organ-specific distant metastases will elucidate prognostic factors. There are drugs that can be used as adjuvant treatment after or during surgery. The use of adjuvant treatment is based on the localized and/or distant metastases, and surgeons will choose the appropriate surgical measures for patients in advance. Based on nomogram scores, there are studies that predict the survival time, thus contributing to prognosis. We currently use the SEER database to determine stages in ovarian cancer and correlate them with survival potential. In the current review, we will present up-to-date information regarding local treatment for ovarian cancer. Cancer stem cells (CSCs) are known to be tumorigenic. They possess the ability to self-reconstruct and initiate tumorigenesis, contrary to normal stem cells [13,14]. Cancer stem cells depend on the tumor type and disease stage [15], and they are quite rare. Moreover, they are responsible for the tumor microenvironment and drug resistance, recurrence, and metastasis [16,17]. They also have the ability to modify the tumor microenvironment. Though it is yet to be elucidated, current findings suggest that the upregulation of cancer stem cells’ pathways is responsible for disease progression [18]. Cancer stem cells have cellular markers that comprise the cell surface and the intracellular, which are connected via a pathway that has close interplay with the tumor microenvironment. 

In many in vivo and in vitro studies, it was suggested that inhibition of cancer stem cell markers effectively downregulates tumor progression [19]. Cancer stem cells were, firstly, observed in ovarian cancer patient ascites [20]. More recently, the correlation of cancer stem cells and ovarian cancer has been extensively investigated. In previously published studies, it was suggested that eliminating cancer stem cells inhibits ovarian cancer growth and frequent disease relapse [19]. Cancer stem cells were previously found in ovarian surfaces and the fallopian tube epithelium [21]. Moreover, cancer stem cells were identified in the coelomic epithelium of mouse ovaries and hilum [22,23,24]. It was also observed that ovarian cancer originates from both these tissues [21,24]. The main crossroad is considered through inactivating TP53 and Rb1genes. These two tumor-suppressor genes are known to mutate in high-grade serous ovarian cancer [25]. Usually, stem cells maintain normal tissue functions; however, dysregulation of their functions induces transformation and tumorigenesis [26]. We collected data from PubMed with references dated until the last five years. In the last five years, only 8 new papers on the current topic were published (Figure 1).

## 2. Morphology and Insights

High-grade serous carcinoma (HGSC) is the most common form of ovarian cancer [27]. High-grade serous carcinoma patients are usually diagnosed at an advanced stage. These patients will receive platinum-based chemotherapy and present good initial responses. However; 75% of these patients will develop platinum resistance within the first five years of treatment. Disseminated from the ovary or fallopian tube, multicellular high-grade serous carcinoma spheroids are spread intraperitoneally within the abdominal cavity lining and, later on, through the lymphatic drainage and systemic circulation to the rest of the body. We also have metastasis to the omentum, which results in tissue transformation composed of adipocytes; this process is also known as desmoplasia or the histological devoiding of adipocytes. The extra cellular matrix in high-grade serous carcinoma presents tumor stiffness, which is also associated with collagen-remodeling signatures and the extension of the desmoplastic area. The higher the stiffness score, the more distant metastasis and, thus, the lower the likelihood of survival. 

Chemotherapy affects the tumor cells, the tumor stroma, and dense stroma fibrosis. Unfortunately, to date, chemotherapy -nduced modifications to the extracellular matrix and cellular responses of high-grade serous carcinoma have not yet been completely identified. Extracellular matrix, tumor microenvironment, systematic circulation, and lymphnode drainage contribute to cancer cells’ ability to resist treatment and metastasize. Extracellular matrix controls cancer cell morphology, survival, invasion, and growth, along with genetic alterations [28]. It is known that the extracellular matrix and tumor microenvironment undergoes major changes in terms of both biomechanical composition and biochemical properties during cancer progression. Moreover, tissue stiffness is also altered. However, we have incomplete information regarding the specific alterations to metastasis and effect of chemotherapy on the extracellular matrix composition [29]. The extracellular matrix has a dynamic and complex molecular network with distinctive structural and biochemical characteristics [30]. It is also known as the matrisome, and it is known to consist of proteins encoded by genes. In the core of extracellular matrix, there are proteins such as proteoglycans, collagens, and ECM glycoproteins, and there are also extracellular matrix-associated proteins, such as ECM-remodeling enzymes and proteins that structurally resemble ECM proteins [31]. If there are structural and mechanical changes, there is tumor promotion and tumor progression [32,33]. External extracellular matrix-mediated stimuli are transduced mainly by integrins [34], which are integrated into focal adhesion complexes, which, in turn, link the cell cytoskeleton to the extracellular matrix to reciprocate forces between cells and the microenvironment [35,36]. In the past few years, several studies were conducted via comprehensive transcriptomics analysis using longitudinal high-grade serous carcinoma cohorts of distinct anatomical sites both pre- and post-chemotherapy. 

To date, we correlated the effects of extracellular matrices with stiffness in cancer cells and chemotherapy responses, observing that treatment-escaping high-grade serous carcinoma cells under platinum treatment induce cell adhesion and migration. These data are also correlated with extracellular matrix protein composition and stiffness. There are published data about tumor-promoting collagen 6 (COL6), which are correlated with the occurrence of chemotherapy-induced changes [37]. 

## 3. Current Data Based on Studies

### 3.1. Conventional Treatment

In the study by Aigner K. et al. [38], an isolated perfusion system was used to overcome drug resistance. In this study, mitomycin and adriamycin was combined in a three-drug regime due to cytotoxicity. In 43% of the patients, a complete decrease in ascites was observed. Toxicity and side effects were mainly grade I and never exceeded grade II. However, post-therapeutic tumor necrosis syndrome was observed. 

In the study by Banik B. et al. [39], it was reported that mitochondria are used as a target for cisplatin. Ovarian cancer cells are sensitive to cisplatin. The mode of action of cisplatin depends on mitophagy and p62 regulation. In the study by Yee S. S. et al. [40], covalent microtubule stabilizers, i.e., the C-22,23-epoxytaccalonolides, were evaluated. Taccalonolide AF was observed to have efficacy in taxane-resistant ovarian cancer models in vitro and in vivo. Moreover, it has unique potential as a intraperitoneal treatment. In the study by Lee H. R. et al. [41], graphene nanoribbons (GNR) speroids with four-arm polyethylene glycol (PEG) and chlorin e6 (Ce6), which are sonosensitizers, were used for metastatic ovarian cancer with mild ultrasound irradiation. An interaction between GNR-PEG and Ce6 was observed with chemotherapeutic agents such as cisplatin and paclitaxel. In the study by Olesen K. D. et al. [42], it was observed that treosulfan could be an alternative when disease relapse occurs in ovarian cancer patients and other treatment options cannot be administered. Treatment was well tolerated by patients with performance statuses in the range 0–1.

### 3.2. Biological Treatment

In the study by Swisher E. et al. [43], ARIEL2, which is a single-arm, open-label phase 2 study of the PARP inhibitor (PARPi) rucaparib in relapsed high-grade ovarian carcinoma with BRCA1/BRCA2 mutations, was performed. In the study by Yin H. et al. [44], the secretory ECM1, which is an isoform that induces tumorigenesis through the G-protein regulatory motif binding to integrin αXβ2 and the activation of AKT/FAK/Rho/cytoskeleton signaling, was investigated. In this study, enhanced sensitivity of cisplatin to cancer cells was observed. These results highlighted extracellular matrix 1a, integrin αXβ2, hnRNPLL, and ATP Binding Cassette Subfamily G Member 1 (ABCG1) as potential targets for treating cancers associated with extracellularmatrix1-activated signaling. 

In the study by Bhojnagarwala P. S. et al. [45], neoantigen tumor-specific antigens were used for ovarian cancer. In this study, up to 40 neoantigens using a single DNA plasmid were targeted. Moreover, vaccines were used to increase T cell responses against multiple heterogeneous tumors as a therapeutic tumor challenge. These vaccines had a long-term immunity. In the case report by Thouvenin L. et al. [46], trastuzumab and pertuzumab was administered, in the case of a young woman with FIGO stage IV high-grade serous ovarian cancer, with an amplification of *ERBB2*. In the study by Larroque M. et al. [47] the HIPEC methodology was used with oxaliplatin for peritoneal metastases (PMs) treatment. The ovary tissue was directly exposed to the drug. This observation was made using LA-ICP MS images. In another case report, a 70-year-old woman had a surgical resection that revealed PLU on pathology report. Based on the pathology of the biopsy and genetic testing, she received entrectinib, but still demonstrated progression in the liver. She was, finally, treated with larotrectinib, which halted disease progression. This example was the first case presented for metastatic PLU involving NRTK3 fusion treated with sequential first-generation NRTK inhibitors [48]. 

In the study by Kang Y. et al. [49], resiquimod was administered, and it enhanced the efficacy of Programmed Death-1 blockade against ovarian tumors. In the study by Wang C. et al. [50], PT@usNLC enhanced the therapeutic effect compared to conventional therapies. In the study by Gonzalez-Juna A. et al. [51], in tumor-bearing immune-competent mice, the murine surrogate of SENTI-101 (mSENTI-101) was administered with local immune response, leading to increased activation of antigen presenting cells, T cells, and B cells. The result was an anti-tumor response and memory-induced long-term immunity. It was also observed that co-administration of mSENTI-101 with checkpoint inhibitors led to synergistic enhancement of anti-tumor response. In the study by Chen Y. et al. [52], immune checkpoint blockade of PD-L1 was administered with LncRNA PVT1-targeted therapy as a double treatment in ovarian cancer patients with cisplatin-resistant recurrence.

### 3.3. Ovarian Cancer and Stem Cells

As was previously shown, ovarian cancer has high heterogeneity due to various factors, such as histopathological properties, clinical evolution, origin, genomic alteration, and response to treatment. It is divided into nine subtypes: serous, endometrioid, mucinous, clear cell, Brenner tumor, squamous cell, transitional cell, mixed epithelial, and the undifferentiated subtype. First-line chemotherapy with a carboplatin–paclitaxel regimen and surgical debulking is the treatment of choice for ovarian cancer [53]. Based on the local and metastatic extent of the disease, the surgeon will choose the adjuvant treatment. It has been observed that the carboplatin–paclitaxel regimen is effective; however, recurrence is observed in ≥70% of patients within the first 5 years after diagnosis, along with acquired chemotherapy resistance [54]. It has been observed that ovarian cancer patients have their first disease relapse within 12 to 18 months [55] due to chemotherapy resistance and metastasis [56]. Overall, 70% of newly diagnosed patients will die within the first 5 years [15,53,57]. 

We urgently need to elucidate factors that are associated with disease recurrence, chemotherapy resistance, and metastasis [58]. Cancer stem cells depend on the tumor type and stage [15], and they are also responsible for drug resistance, metastasis, and recurrence. Their behavior depends on the tumor microenvironmental or non-microenvironmental stimuli [16]. It has been observed that cancer stem cells are responsible for the poor treatment outcomes for many aggressive tumors [18]. Cancer stem cells have cellular markers on their cell surface and within cells. These markers are closely connected to the tumor microenvironment’s behavior. In previous studies, it was reported that if these markers are inhibited individually or combined, cancer stem cell growth stops both in vitro and in vivo [59]. Cancer stem cells were first observed in ovarian cancer patient ascites [20]; since then, there has been growing investigation of the correlation between cancer stem cells and ovarian cancer. The elimination of cancer stem cells inhibits ovarian cancer growth and blocks disease recurrence [19,59]. In several studies, stem-like epithelial cells were identified in the ovarian surface [21,60] and fallopian tube epithelium [60]. Additionally, these cells were found in the coelomic epithelium of mouse ovaries [22,23] and in hilum [23]. Furthermore, it was observed that ovarian cancer originates from coelonic and hilum [19,24]. Moreover, it was observed that TP53 and Rb1 are frequently mutated in high-grade serous ovarian cancer [25]. Stem cells maintain the normal regeneration of tissues; however, their dysregulation and transformation can cause cancer [26]. In the study by Mansoori M. et al. [61], it was observed that increased expression of GD2 can predict treatment and be used as a diagnostic marker (Table 1).

## 4. Discussion

It is known that for patients with low residual disease, the risk of recurrence ranges from 65 to 75%. It was observed that after surgery, the risk of recurrence is about 85–90% in women with large-volume residual disease. Upon diagnosis, patients are usually at an advanced stage and, therefore, debulking surgery is performed and followed by platinum-based adjuvant treatment. Roughly 60–70% of patients with advanced disease relapse and chemotherapy resistance is observed. Finally, salvage chemotherapy is administered. In order to decide on the optimal chemotherapy treatment, we need to know the time period of treatment administration. Usually, patients with recurrent ovarian cancer have platinum-sensitive or platinum-resistant disease. The Fifth Ovarian Cancer Consensus suggested distinguishing between platinum-sensitive and platinum-resistant patients [62]. To date, the classic classification is still used [63]. Patients are considered platinum insensitive in cases where the time-free interval is >6 months, while patients with a time free interval between 6 and 12 months are considered partially platinum sensitive, and patients with a time free interval >12 months are considered completely platinum sensitive. In the case that the time free interval is equal to 6 months, patients are deemed platinum resistant. Time-free interval is a reliable predictor of response to second-line chemotherapy. Platinum-based combination chemotherapies are recommended as treatment for patients with platinum-sensitive ovarian cancer [64]. The longer time-free interval was observed to be associated with higher overall survival. Specifically, if the response rates for time-free interval is 5–12 months, 13–24 months, or >24 months, survival rates are 27%, 33%, and 59%, respectively. Extension of platinum-free interval with a non-platinum-based regimen improves survival outcomes [65]. Platinum agents alone or combined with paclitaxel, liposomal doxorubicin or gemcitabine, or topotecan or liposomal doxorubicin, are options for partially platinum-sensitive patients with time-free intervals between 6 and 12 months [65,66]. There are very few studies comparing treatment outcomes for patients with partial platinum sensitivity and patients with platinum-sensitive recurrent ovarian cancer. 

In the study of Pujade-Lauraine et al., carboplatin and liposomal doxorubicin combination was superior to carboplatin and paclitaxel in patients with platinum-sensitive recurrent ovarian cancer [67]. These prior studies, although well-designed, had results that were not yet fully elucidated. Furthermore, platinum–paclitaxel combination had higher OS than platinum with liposomal doxorubicin in patients with a recurrent disease-free interval between six and twelve months. The same observation was made for platinum with paclitaxel, as well as for platinum with liposomal doxorubicin, in patients with time-free intervals of >12 months.

Several studies presented data where ovarian cancer patients with longer platinum-free intervals had higher OS in recurrent ovarian cancer [68]. It has not yet been elucidated whether extending the platinum-free interval with non-platinum agents improves the response after re-introducing platinum-based chemotherapy. The MITO8 phase III study was one of the studies that investigated this treatment model, using longer platinum-free intervals in patients with ovarian cancer recurring between 6 and 12 months after previous platinum-based chemotherapy. In another study, the median OS values were investigated for non-platinum- and platinum-based chemotherapeutic groups [69]. In an Asian population that had time-free intervals between 6 and 12 months and were treated with non-platinum-based regimens, poorer outcomes were observed than in those patients directly re-treated with platinum-based regimens. However, there were several issues that influenced the results of MITO-8, including the initial stages reported, histological types, different ethnicities, and statuses of primary or secondary debulking surgeries. It was observed that the five-year overall survival (OR) rate was poor in a 5-year observation period for patients treated with topotecan or pegylated liposomal doxorubicin alone compared to the platinum therapy group. Furthermore, it was observed that the platinum-based doublet regimens with topotecan or doxorubicin had a 5-year OS that was higher than the value for non-platinum single regimens using either topotecan or doxorubicin alone. The CALYPSO trial revealed lower overall survival rates in patients treated with platinum topotecan than in the platinum doxorubicin groups [70]. It was observed that the platinum and doxorubicin group experienced reduced toxicity compared to the platinum topotecan group [67]. Liposomal doxorubicin has been recommended as an addition to platinum instead of paclitaxel for those patients with recurrent ovarian cancer and time-free intervals of 6 to 12 months [67,71]. In the CALYPSO trial, patients in the platinum and doxorubicin and platinum and topotecan groups did not complete six cycles of chemotherapy [67]. All patients that received at least six cycles of chemotherapy had a higher OS. All patients that received more than six cycles of chemotherapy had a poorer performance status. The different drugs led to different results between studies of Caelyx (Eli Lily, Bruxelles, Belgium) [67,72] and Lipo-Dox (TTY BioPharm, Taipei City, Taiwan) [73,74]. Another issue for different overall survival rates is the different populations included, i.e., Caucasians and Asians. Moreover, different doses were used in different studies and, therefore, different adverse effects were observed in front-line ovarian cancer treatment [75]. Furthermore, in another study, pazopanib was used as a maintenance treatment in ovarian cancer [76]. 

Currently platinum and topotecan is recommended as the first-choice treatment for patients with recurrent ovarian cancer with a time-free interval of 6 to 12 months. Platinum and doxorubicin is considered only for patients with intolerable paclitaxel-related toxicities [77]. Platinum-based combination chemotherapy with paclitaxel or liposomal doxorubicin had similar OS rates for patients with recurrent ovarian cancer with time-free intervals of >12 months [65,70]. In the CALYPSO trial, the platinum–topotecan groups had significantly higher incidences of grade 3–4 neutropenia, grade 2 alopecia, grade 2 sensory neuropathy, allergic reaction, and arthralgia/myalgia compared to the PD group [67]. Platinum–doxorubicin is recommended as the first-choice chemotherapy regimen for patients with recurrent ovarian cancer with time-free intervals of >12 months because platinum–doxorubicin regimens had similar outcomes and fewer toxicities. 

The platinum-sensitive patient category changed during the past 10 years [78,79], and another three studies related to the maintenance of PARP inhibitors [80] were published. There are published data indicating benefits in terms of progression-free survival. In several studies, we observed that PARP inhibitors accounted for a significant improvement in progression-free survival in both recurrent and primary ovarian cancer patients, regardless of patients’ breast cancer gene mutational status [81]. Radiation therapy has a role as localized treatment to treat ovarian cancer patients with locally recurrent vaginal or perirectal lesions. The role of this treatment modality remains unclear, though we hypothesize that these patients may be treated locally and gain long-term survival benefits [82]. 

## 5. Conclusions

Currently, we are still trying to investigate in depth the intracellular mechanisms related to ovarian cancer, along with the tumor micro-environment and immunosuppressive drugs. We are also trying to add anti-angiogenetic agents, which can enhance the treatment by changing the tumor stroma and blocking chemotherapy resistance. Unfortunately, these mechanisms are multifactorial, and the same drugs that we administer with time during treatment produce different cancer stem clones that are chemotherapy resistant. The cancer stem cells evolve over time, with this process actually being detected during the clinical trial. We need to develop new treatment strategies and administer dual–triple therapeutic modalities to overcome chemotherapy resistance. It is evident that targeting only one mechanism is futile. We must produce a personalized treatment model after the first-line treatment based on several molecular characteristics of a patients, which are not clear even after years of research. We need to use current biomarkers to properly address the issue of chemotherapy resistance and direct more patients to different clinical trials. We also need to properly identify and successfully validate each patient in order to predict the risk of individual patients developing resistance to Pt-BMs.

## Figures and Tables

**Figure 1 medicina-59-01183-f001:**
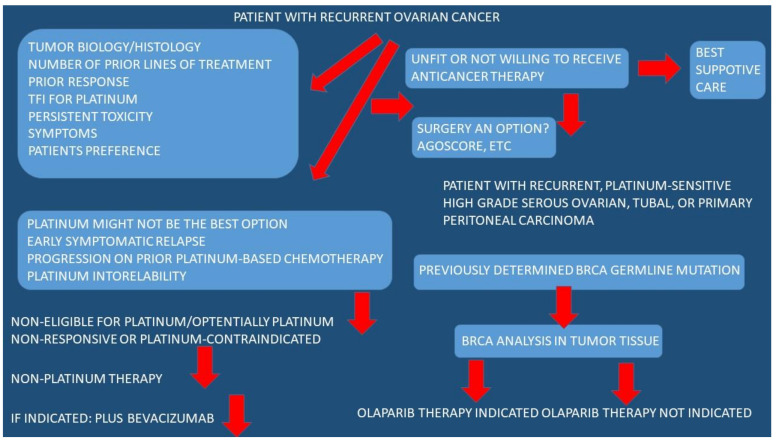
It is known that ovarian cancer is often diagnosed at a late stage. Stem cell dysfunction or reproduction dysregulation is the main reason for ovarian cancer growth. Surgery and chemotherapy is possible for early-stage disease; however, 60% of the patients experience disease relapse during the following 12–18 months. Finally chemotherapy resistance is observed in 80–90% of all patients.

**Table 1 medicina-59-01183-t001:** Ongoing clinical trials.

Study	Therapeutic Target	Condition	Trial Type	Phase	Status	Identifier
JGOG-3017	-	Stage I–IV	Randomized	3	Completed	-
GOG-268	mTOR	Stage III–IV	Single Group	2	Completed	NCT01196429
GOG-254	VEGFR, PDGFR	Persistant–Recurrent	Single Group	2	Completed	NCT00979992
ENMD-2076	Aurora A, VEGFR, FGFR	Persistant–Recurrent	Single Group	2	Completed	NCT01104675
NiCCC	VEGFR, PDGFR, FGFR	Persistant–Recurrent	Randomized	2	Ongoing	NCT02866370
NRG-GY001	MET, RET, VEGFR2	Persistant–Recurrent	Single Group	2	Completed	NCT02315430
GOG-283	ABL, Src, c-kit	Persistant–Recurrent	Single Group	2	Completed	NCT02059265
MOCCA	PD-L1	Persistant–Recurrent	Randomized	2	Ongoing	NCT03405454
BrUOG-354	PD-1, CTLA-4	Persistant–Recurrent	Randomized	2	Ongoing	NCT03355976

CTLA-4; cytotoxic T-lymphocyte-associated protein 4, GOG; Gynecologic Oncology Group; FGFR; fibroblast growth factor receptor, JGOG; Japanese Gynecologic Oncology Group, MOCCA; multicentre phase II trial of durvalumab versus physician’s choice chemotherapy in recurrent ovarian clear cell adenocarcinomas, mTOR; mammalian target of rapamycin, BrUOG; Brown University Oncology Research Group, NiCCC; study of nintetanib compared to chemotherapy in patients with recurrent clear cell carcinoma of the ovary or endometrium, PD-1; programmed cell death-1, PDGFR; platelet-derived growth factor receptor, PD-L1; programmed death ligand-1, VEGFR; vascular endothelial growth factor receptor.

## Data Availability

The data presented in this study are available on request from the corresponding author.

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
