# Peer review of "Epithelial Ovarian Cancer: A Five Year Review"

_medicina, 2023, doi:10.3390/medicina59071183_

Round 1

Reviewer 1 Report

The manuscript included much information regarding epithelial ovarian cancer morphology, molecular biology, and treatment options. The content was presented in long paragraphs and it is difficult to understand various points, with or without context. If break the content into multiple paragraphs, it would be easier to understand. 

Figure 2 and table 1 are redundant; column #5 has no title. 

Figure 3 demonstrated ongoing clinical trials for clear cell carcinoma, but there was no context. 

Minor editing in language is needed.

Author Response

The manuscript included much information regarding epithelial ovarian cancer morphology, molecular biology, and treatment options. The content was presented in long paragraphs and it is difficult to understand various points, with or without context. If break the content into multiple paragraphs, it would be easier to understand. 

Thank you for your comment

Answer

We have divided the manuscript into more paragraphs

Figure 2 and table 1 are redundant; column #5 has no title. 

Thank you for your comment

Answer

We have removed Figure number 2 and Table 1 as indicated by you.

Figure 3 demonstrated ongoing clinical trials for clear cell carcinoma, but there was no context. 

Thank you for your comment

Answer

We do not have any additional data regarding these trials and that is why we did not include more information

Reviewer 2 Report

Main contents and dicussion is not associated well.

Main contents describes cancer stem cell and genetic alterations and anothers.

But discussion describes only about chemotherapy.

In my opinion, discussion may include and summarize main contents.

In discussion, 'Oss' must be corrected and full name of 'OR' described.

Author Response

Reviewer 2

Comments and Suggestions for Authors

Main contents and dicussion is not associated well.

Main contents describes cancer stem cell and genetic alterations and anothers.

But discussion describes only about chemotherapy.

In my opinion, discussion may include and summarize main contents.

Thank you for your comments

Answer

We have divided the manuscript into more paragraphs as indicated by reviewer number 1 and we have erased figure 2 and Table 1. As indicated by reviewer number 1

We have also added a summary in the discussion section as indicated by you

Comments on the Quality of English Language

In discussion, 'Oss' must be corrected and full name of 'OR' described.

The `Oss` has been corrected to `OS`

And `OR` has been corrected to `Overall Survival`

Round 2

Reviewer 1 Report

In "Current data based on studies", many aspects of biology and therapy were discussed. However, the topics were not closely related and were grouped in one paragraph. Recommend break the contents into more paragraphs.  

Minor issues, such as punctuations. 

Author Response

Reviewer 1

In "Current data based on studies", many aspects of biology and therapy were discussed. However, the topics were not closely related and were grouped in one paragraph. Recommend break the contents into more paragraphs.  

Thank you for your comment

Answer

We chose to divide the Paragraph into studies related with conventional treatment and biological treatment

Comments on the Quality of English Language

Minor issues, such as punctuations. 

Thank you for your comment

Answer

We performed corrections where necessary

Reviewer 2 Report

Your revison is suitable to accept.

Author Response

Reviewer 2

Comments and Suggestions for Authors

Your revison is suitable to accept.

Thank you for your comment

Answer

Thank you
